# CM-GAN: Enhancing Consistency Model Image Quality and Stabilizing GAN Training

## Abstract

Generative adversarial networks (GANs) have gained significant attention for generating realistic images, but they are notoriously difficult to train. In contrast, diffusion models provide stable training and avoid mode collapse, though their generation process is computationally intensive. To address this, Song et al. (2023) introduced consistency models (CMs), which optimize a novel consistency constraint derived from diffusion processes. In this paper, we propose a training method, CM-GAN, combining the strengths of both diffusion models and GANs while overcoming their respective limitations. We demonstrate that the same consistency constraint can be applied to stabilize GAN training and mitigate mode collapse. Meanwhile, CM-GAN serves as a fine-tuning mechanism for CMs by leveraging a discriminator, resulting in superior performance compared to CMs alone. Empirical results on benchmarks such as ImageNet $64{\times}64$ and Bedroom $256{\times}256$ show that CM-GAN significantly enhances the sample quality of CMs and effectively stabilizes GAN training.

## 1 Introduction

Generative adversarial networks (Goodfellow et al., 2014; Brock et al., 2019; Karras et al., 2021b) have made remarkable success in generating high-resolution images that closely resemble real photos. However, practical implementation of generative adversarial networks (GANs) often encounters several challenges, such as non-convergence, training instability, and mode collapse, where the generated outputs become repetitive or limited in variation (Goodfellow, 2016; Arjovsky & Bottou, 2017; Mescheder et al., 2018). To address these challenges, many theoretical and empirical attempts have been made including: enhancing network architectures (Mescheder et al., 2017; Arjovsky & Bottou, 2017; Li et al., 2017a), developing theoretical insights into GAN training dynamics (Nowozin et al., 2016), devising new objective functions (Nowozin et al., 2016; Arjovsky et al., 2017; Zheng & Zhou, 2021), and incorporating mappings from data to latent representations (Donahue et al., 2017; Dumoulin et al., 2017; Li et al., 2017b).

Recently, diffusion-based generative models (Sohl-Dickstein et al., 2015; Ho et al., 2020; Song et al., 2021a;b; 2023) have gained increasing attention and many impressive breakthroughs have been made (Croitoru et al., 2023) in generating images (Ho et al., 2020; Song et al., 2021a;b; Rombach et al., 2022; Song et al., 2023), audios (Kong et al., 2021; Yang et al., 2023) and videos (Ho et al., 2022). Due to some inherent properties, diffusion models are relatively easier to train and do not suffer from those common training difficulties of GANs. In contrast, its generation process involves iteratively applying denoising steps to progressively transform noise into data samples (Ho et al., 2020) or solving a complex ODE system using an iterative solver (Song et al., 2021b), which is computationally expensive. To alleviate this difficulty, Song et al. (2023) proposed consistency models (CMs). By adopting a novel local consistency constraint, the model can be either distilled from a pre-trained diffusion model or trained from scratch, enabling a single-step generation process.

In this paper, we introduce a novel approach that leverages consistency constraints to enhance GAN training stability and address the issue of mode collapse. Our method uses an under-trained diffusion model as a prototype, enforcing consistency to ensure the generator's outputs align with those of the diffusion model.

This approach combines the strengths of both diffusion models and GANs while mitigating their key weaknesses. Additionally, it acts as a fine-tuning mechanism for CMs by integrating a discriminator, potentially exceeding the performance of standard CMs. Empirical results on ImageNet 64×64 and Bedroom 256×256 demonstrate that CM-GAN significantly improves sample quality and stabilizes GAN training.

## 2 Related work

Diffusion models have demonstrated impressive capabilities in generating and editing high-resolution images (Balaji et al., 2023; Ramesh et al., 2022; Rombach et al., 2022) and videos (Ho et al., 2022; Blattmann et al., 2023), but their iterative nature poses challenges for real-time use.

Latent diffusion models (Rombach et al., 2022) attempt to solve this problem by representing images in a more computationally feasible latent space (Esser et al., 2020). However, they still rely on the iterative application of large models with billions of parameters.

Alongside the development of faster samplers for diffusion models (Song et al., 2021a; Lu et al., 2022; Dockhorn et al., 2022; Zheng et al., 2023), there is increasing interest in model distillation techniques such as progressive distillation (Salimans & Ho, 2022) and guidance distillation (Meng et al., 2023). These methods can reduce the number of iterative sampling steps to as few as 4-8; however, they often lead to a noticeable decline in sample quality and demand a labour-intensive iterative training process.

Consistency models (Song et al., 2023; Song & Dhariwal, 2024) address performance degradation by enforcing consistency regularization on the ODE trajectory, delivering solid results in few-shot settings for pixel-based models. Specifically, in diffusion models, the PF-ODE trajectory has two key points: the start (a data sample) and the endpoint (Gaussian noise). Consistency models (CMs) are trained to predict the data sample from any point along this trajectory. In particular, when provided with Gaussian noise, CMs can directly return the corresponding data sample, allowing for efficient one-step sampling. Building on this framework, Kim et al. (2023b) extended the original CM setting by training a model that can predict any point along the trajectory from any other point with a single function evaluation. Furthermore, Luo et al. (2023a) concentrated on distilling latent diffusion models with consistency constraints, achieving impressive performance with only four sampling steps. In follow-up work, LCM-LoRA (Luo et al., 2023b) introduced a low-rank adaptation (Hu et al., 2021) technique that enables efficient training of LCM modules. While these methods train CM models using consistency loss (see Section 3.2 for details), which requires the model to produce the same output for any two consecutive points on the same PF-ODE trajectory, Kang et al. (2025) instead train the model to directly predict the start point from the endpoint, bypassing intermediate trajectory points. This simplification increases training difficulty, which the authors addressed by initializing the model from a pre-trained diffusion model and incorporating adversarial training objectives to enhance performance.

Another method to avoid iterative sampling is Rectified Flows (Liu et al., 2022), which dynamically adjusts the data-noise coupling, effectively linearizing the ODE path and significantly reducing the number of iterations to solve it. This approach was incorporated into InstaFlow (Liu et al., 2023), enabling one-step latent-space sampling for text-to-image generation tasks. Although these methods reduce the number of iterations, they often degrade the quality of the generated samples, particularly when reducing the steps to just one or two.

Generative Adversarial Networks (GANs) represent another prominent category of generative models (Goodfellow et al., 2014). These models can also be implemented as independent, single-step systems for converting text into images similar to latent diffusions (Sauer et al., 2023a; Kang et al., 2023). While GANs excel in rapid image generation, their overall quality often falls short of diffusion-based approaches. This performance gap may stem from the complex GAN-specific architectures required to maintain stability during adversarial training, making improvements without destabilizing the model challenging. For instance, Karras et al. (2018) proposed a progressive generator architecture that trains high-resolution generators by starting at low resolution and gradually increasing it. While this approach improves training stability, the resulting image quality still lags behind state-of-the-art diffusion models, such as EDM (Karras et al., 2022). Furthermore, unlike large-scale diffusion models, leading text-to-image GANs lack an equivalent to classifier-free

guidance (Ho & Salimans, 2022), a key post-training technique that can significantly enhance output quality through score-based continuous sampling – a sampling method exclusive to diffusion models.

Score Distillation Sampling (SDS, Poole et al. 2022), also known as Score Jacobian Chaining (Wang et al., 2023), is a recently introduced technique for transferring knowledge from large-scale text-to-image models to 3D synthesis models. Recent research has shown that score-based models are closely connected to GANs (Franceschi et al., 2023), which has inspired the development of Score GANs—models that use score-based diffusion flows from a Diffusion Model (DM) instead of a traditional discriminator for training. Building on the principles of SDS, Diff-Instruct (Luo et al., 2023c) extends this approach by enabling the distillation of a pre-trained diffusion model into a generator without the need for a discriminator.

Concurrently, researchers are exploring ways to enhance the diffusion process through adversarial training. For instance, Denoising Diffusion GANs (Xiao et al., 2022) have been introduced to enable rapid sampling with fewer steps. To improve output quality, Kim et al. (2023a) introduced a discriminator that evaluates the realisability of a denoising sampling path. The discriminator will then be used to correct the sampling trajectory and improve the image quality. In a closely related setting (Song & Ermon, 2020) that combines annealed Langevin sampling (Welling & Teh, 2011; Roberts & Tweedie, 1996) and Denoising Score Matching (Hyvärinen, 2005), Jolicoeur-Martineau et al. (2020) rewrite the score estimation task as a denoising task and uses a discriminator to encourage the denoised image to be realistic for all noise levels. A similar idea is later adopted by Sauer et al. (2023b) in the distillation of latent diffusion models using a generalized SDS. In their work, the auxiliary adversarial training objective is instead applied to the denoising task for all time $t$. For stabler gradients, they implement distillation loss in the pixel space, which is discarded in the follow-up work through a simplified training pipeline (Sauer et al., 2024).

## 3 Preliminary

### 3.1 Generative adversarial networks

Generative adversarial networks (GAN, Goodfellow et al. 2014) are a family of generative models that learn a data distribution $p_{\text{data}}$ by establishing a min-max game between two neural networks: a generator $\mathcal{G}$ and a discriminator $\mathcal{D}$.

The generator $\mathcal{G}$ takes a random noise vector $\mathbf{z}$ sampled from a prior distribution $p_{\text{prior}}$ (typically a spherical Gaussian) and outputs a generated (fake) sample $\mathbf{y} = \mathcal{G}(\mathbf{z}) \sim p_{\mathcal{G}}$. Meanwhile, the discriminator $\mathcal{D}$ is trained to distinguish between fake samples $\mathbf{y}$ and real data $\mathbf{x}$. Specifically, $\mathcal{D}$ is optimized to correctly classify real training samples from $p_{\text{data}}$ and the fake samples generated by $\mathcal{G}$, while $\mathcal{G}$ is trained to generate more realistic samples that can fool $\mathcal{D}$. This adversarial dynamic is captured by the following min-max objective function:

$$\min_{\mathcal{G}} \max_{\mathcal{D}} \ \mathcal{L}_{\text{GAN}}(p_{\mathcal{G}}, \mathcal{D}). \tag{1}$$

In the original GAN formulation, $\mathcal{L}_{\text{GAN}}$ is defined as:

$$\mathcal{L}_{\text{GAN}}(p_{\mathcal{G}}, \mathcal{D}) = \mathbb{E}_{\mathbf{x} \sim p_{\text{data}}}\Big[ \log \mathcal{D}(\mathbf{x}) \Big] - \mathbb{E}_{\mathbf{y} \sim p_{\mathcal{G}}}\Big[ -\log\big(1 - \mathcal{D}(\mathbf{y})\big) \Big]. \tag{2}$$

However, optimizing GANs is often unstable and can suffer from the gradient vanishing problem. As a result, various modifications to the objective function Eq (2) have been proposed to improve the stability and performance of GANs (Goodfellow et al., 2014; Arjovsky et al., 2017; Miyato et al., 2018; Fedus et al., 2018), though the underlying adversarial dynamic between $\mathcal{G}$ and $\mathcal{D}$ remains unchanged.

For many GAN variants, $\mathcal{L}_{\text{GAN}}(p_{\mathcal{G}}, \mathcal{D})$ can be generalized as the difference between two expectations (Nowozin et al., 2016):

$$\mathcal{L}_{\text{GAN}}(p_{\mathcal{G}}, \mathcal{D}) = \mathbb{E}_{\mathbf{x} \sim p_{\text{data}}} \psi_1(\mathcal{D}(\mathbf{x})) - \mathbb{E}_{\mathbf{y} \sim p_{\mathcal{G}}} \psi_2(\mathcal{D}(\mathbf{y})), \tag{3}$$

where $\psi_1$ and $\psi_2$ are functions that depend on the specific GAN variant being used. In the original GAN formulation, $\psi_1(\mathbf{z}) = \log(\mathbf{z})$ and $\psi_2(\mathbf{z}) = -\log(1 - \mathbf{z})$.

Another common problem in GANs is mode collapse, where the generator barely produces a small set of outputs (Goodfellow, 2016; Arjovsky & Bottou, 2017; Mescheder et al., 2018). This happens because the generator $\mathcal{G}$ is trained to find the output that seems most plausible to the discriminator. Once $\mathcal{G}$ starts generating the same output (or a small set of outputs) consistently, the discriminator $\mathcal{D}$ may choose to remember this output and always reject it, which could get $\mathcal{D}$ stuck at a local optimum. As a result, for the next iteration, $\mathcal{G}$ could find the most plausible output for $\mathcal{D}$ easily while $\mathcal{D}$ fails to effectively improve its learning to escape this predicament. Consequently, the generator and discriminator end up cycling through a limited range of outputs.

In Section 4, we will show that the challenges mentioned earlier can be significantly mitigated by incorporating the consistency constraint (Song et al., 2023). This constraint is enforced by leveraging a pretrained diffusion model as a "prior" model, ensuring that the generator $\mathcal{G}$ remains in proximity to the prior and consistently generates diverse outputs. Thus, training becomes more stable and mode collapse is effectively avoided.

### 3.2 Probability flow ODE and consistency models

The probability Flow (PF) ODE and consistency models (CMs) are two families of generative models that are closely related to the continuous-time diffusion models (Song et al., 2021b). Diffusion models generate data by iteratively introducing Gaussian perturbations to the input data, gradually transforming it into noise, and subsequently generating samples from the noise through a series of sequential denoising steps. Given data distribution $p_{\text{data}}$, the forward perturbation is characterized by a stochastic differential equation:

$$\mathrm{d}\mathbf{x}_t = \boldsymbol{\mu}(\mathbf{x}_t, t)\,\mathrm{d}t + \sigma(t)\,\mathrm{d}\mathbf{w}_t, \tag{4}$$

for $t \in [0, T]$ and $T$ is a fixed positive constant. $\boldsymbol{\mu}(\cdot, \cdot)$ and $\sigma(t)$ denote the drift and diffusion coefficients while $\{\mathbf{w}_t\}_{t \in [0,T]}$ is the standard Brownian motion. In this paper, we adopt the same configuration as Song et al.'s, where $\boldsymbol{\mu}(\mathbf{x}, t) = 0$ and $\sigma(t) = \sqrt{2t}$. When $T$ is sufficiently large, $\mathbf{x}_T$ can be approximately seen as a sample following $\mathcal{N}(\mathbf{0}, T^2\mathbf{I})$. Let $p_t$ denote the distribution of $\mathbf{x}_t$ (thus, $p_0 = p_{\text{data}}$ and $p_T \approx \mathcal{N}(\mathbf{0}, T^2\mathbf{I})$). Song et al. (2021b) proved that the solution $\tilde{\mathbf{x}}_t$ of the ODE:

$$\mathrm{d}\tilde{\mathbf{x}}_t = \left[ -t\,\nabla \log p_t(\tilde{\mathbf{x}}_t) \right] \mathrm{d}t \quad \text{with } \tilde{\mathbf{x}}_T \sim p_T(\tilde{\mathbf{x}}_T) \tag{5}$$

is also distributed according to $p_t$, where the ODE in Eq (5) is called the *PF-ODE*. Here, $\nabla \log p_t(\mathbf{x}_t)$ is the score function of $p_t(\mathbf{x}_t)$ and can be empirically estimated by a neural network $\mathbf{s}_\phi(\mathbf{x}_t, t)$ which is notably easy to train due to the stable training process. (Readers may refer to Song et al. (2021b) for its training details.) With a well-trained $\mathbf{s}_\phi(\mathbf{x}_t, t)$, we then can plug it into Eq (5) and solve the PF-ODE backward starting from $\tilde{\mathbf{x}}_T \sim \mathcal{N}(\mathbf{0}, T^2\mathbf{I})$ and the resulting $\tilde{\mathbf{x}}_0$ can be seen as an approximate sample of $p_{\text{data}}$.

Solving PF-ODE is generally expensive, which motivates Song et al. (2023) to propose CMs. Specifically, they train a neural network $\mathbf{f}_{\boldsymbol{\theta}}(\mathbf{x}_t, t)$ that maps any point $(\mathbf{x}_t, t)$ on the PF-ODE trajectory to its origin $(\mathbf{x}_0, 0)$. Then for $\tilde{\mathbf{x}}_T \sim \mathcal{N}(\mathbf{0}, T^2\mathbf{I})$, $\mathbf{f}_{\boldsymbol{\theta}}(\mathbf{x}_T, T)$ is an approximate sample of $p_{\text{data}}$ and the iterative ODE solving process is avoided. To train $\mathbf{f}_{\boldsymbol{\theta}}$, they discretize interval $[0, T]$ into $N-1$ sub-intervals with boundaries $0 = t_1 < t_2 < \cdots < t_N = T$ and adopt a special model architecture so that $\mathbf{f}_{\boldsymbol{\theta}}(\mathbf{x}_0, 0) = \mathbf{x}_0$. Then $\mathbf{f}_{\boldsymbol{\theta}}$ is trained to minimize a consistency distillation loss:

$$\mathcal{L}_{\text{CD}}(\boldsymbol{\theta}, \bar{\boldsymbol{\theta}}) = \mathbb{E}\left[ \left\| \mathbf{f}_{\boldsymbol{\theta}}(\mathbf{x}_{t_{n+1}}, t_{n+1}) - \mathbf{f}_{\bar{\boldsymbol{\theta}}}(\hat{\mathbf{x}}_{t_n}, t_n) \right\|_2^2 \right] \tag{6}$$

where expectation is taken with respect to $\mathbf{x} \sim p_{\text{data}}$, $n \sim \mathcal{U}[\![1, N-1]\!]$, $\mathbf{x}_{t_{n+1}} \sim \mathcal{N}(\mathbf{x}; t_{n+1}^2\mathbf{I})$. Here, $\mathcal{U}[\![1, N-1]\!]$ denotes a uniform distribution over $\{1, 2, \cdots, N-1\}$. $\hat{\mathbf{x}}_{t_n}$ is the solution at step $t_n$ of the PF-ODE trajectory through $(\mathbf{x}_{t_{n+1}}, t_{n+1})$ and can be estimated through an Euler method starting from $(\mathbf{x}_{t_{n+1}}, t_{n+1})$ with a pre-trained $\mathbf{s}_\phi$. Specifically,

$$\hat{\mathbf{x}}_{t_n} \approx \mathbf{x}_{t_{n+1}} - (t_n - t_{n+1})\,t_{n+1}\mathbf{s}_\phi(\mathbf{x}_{t_{n+1}}, t_{n+1}).$$

In addition, $\bar{\boldsymbol{\theta}}$ denotes a running average of the past values of $\boldsymbol{\theta}$.

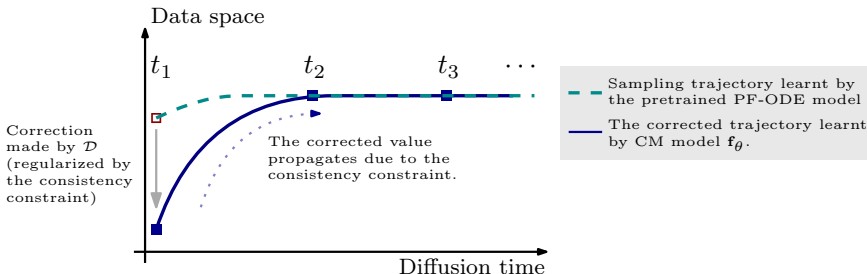

Figure 1: Discriminator $\mathcal{D}$ corrects the outputs of the generator, CM model $\mathbf{f_\theta}$, while the consistency constraint ensures that the corrected output stays close to the one induced by the PF-ODE.

To understand why $\mathcal{L}_{\mathrm{CD}}$ is effective, assume that $\mathbf{f_\theta}$ is well-trained, and that $\mathbf{f_\theta}(\mathbf{x}_{t_n}, t_n) = \mathbf{f_\theta}(\mathbf{x}_{t_{n+1}}, t_{n+1})$ for all $n = 1, 2, \ldots, N - 1$. By recursively applying this equality from $t_1$, we get $\mathbf{x}_0 = \mathbf{f_\theta}(\mathbf{x}_{t_1}, t_1) = \cdots = \mathbf{f_\theta}(\mathbf{x}_{t_N}, t_N)$. Therefore, by minimizing $\mathcal{L}_{\mathrm{CD}}$, $\mathbf{f_\theta}(\mathbf{x}, t)$ is trained to return the origin $\mathbf{x}_0$ of the PF-ODE trajectory for all $(\mathbf{x}, t)$ along the trajectory.

Then by sampling $\tilde{\mathbf{x}}_T \sim \mathcal{N}(\mathbf{0}, T^2\mathbf{I})$ and evaluating $\mathbf{f_\theta}(\mathbf{x}_T, T)$, CM generates an approximate sample $\hat{\mathbf{x}}_0$ from $p_{\mathrm{data}}$ in a single step. To enhance sample quality, practitioners may add noise $\boldsymbol{\epsilon} \sim \mathcal{N}(0, \mathbf{I})$ to $\hat{\mathbf{x}}_0$, yielding $\hat{\mathbf{x}}_t = \hat{\mathbf{x}}_0 + t\boldsymbol{\epsilon}$ for $t < T$, and then evaluate $\mathbf{f_\theta}(\hat{\mathbf{x}}_t, t)$ again. This process can be iteratively repeated to improve image quality, though the marginal improvements diminish with each additional function evaluation.

We would like to note that fast sampling of CMs comes with a trade-off in output quality since the pre-trained PF-ODE model cannot be perfectly distilled in general. Additionally, the performance of CMs heavily depends on the quality of the pre-trained PF-ODE model, emphasizing the significance of a well-trained model for achieving desirable results. In the subsequent section, we will demonstrate that the performance of CMs can be enhanced by incorporating an adversarial training setting. This approach not only improves the performance of CMs but also alleviates concerns regarding the imperfect training of the PF-ODE model.

## 4 Approach

In this section, we introduce a method that can serve as both a technique to enhance the training stability of GANs and improve the performance of CMs. The approach assumes the accessibility to a pre-trained PF-ODE model (not necessarily to be perfectly trained), which will serve as a prototype of the generator $\mathcal{G}$ (from the view of stabilizing GAN's training) or the model to be distilled (from the view of enhancing CMs).[1] To emphasize the reliance on the consistency constraint in CMs, we name our approach CM-GAN. We will begin by presenting our method as a fine-tuning technique for CMs, which provides a clearer understanding and stronger motivation for our work.

Consider the distillation process of CMs that minimizes $\mathcal{L}_{\mathrm{CD}}$ in Eq (6). Due to a possibly imperfect training of CM and the pretrained PF-ODE model, $\mathbf{f_\theta}$ could not output a good enough approximate sample of $p_{\mathrm{data}}$. To fix this issue, we can adopt a GAN structure by simultaneously training a discriminator $\mathcal{D}$ to correct the outputs of $\mathbf{f_\theta}(\mathbf{x}_{t_n}, t_n)$ for $\mathbf{x}_{t_n} \sim p_t(\mathbf{x}_{t_n})$ and $n \sim \mathcal{U}[\![1, N - 1]\!]$. In this way, the error signal from $\mathcal{D}$ guides $\mathbf{f_\theta}$ to produce more realistic outputs while the consistency constraints regularize the corrected output to stay in the neighbour of the one induced by the PF-ODE (the ground truth in the distillation of CMs).

To see how discriminator $\mathcal{D}$ helps the training of $\mathbf{f_\theta}$, consider the training dynamic involving the time step $t_1 = 0$ (see Fig 1). The consistency constraint $\|\mathbf{f_\theta}(\mathbf{x}_{t_2}, t_2) - \mathbf{f_{\bar\theta}}(\mathbf{x}_{t_1}, t_1)\|^2$ enforces $\mathbf{f_\theta}(\mathbf{x}_{t_2}, t_2)$ to stay close to the origin of the PF-ODE trajectory $\mathbf{f_\theta}(\mathbf{x}_{t_1}, t_1)$ while $\mathcal{D}$ provides additional correction signal to make $\mathbf{f_\theta}(\mathbf{x}_{t_2}, t_2)$ be more realistic. We apply this idea recursively and obtain the following training objective:

$$\min_{\mathbf{f_\theta}} \max_{\mathcal{D}_\phi} \ \mathcal{L}_{\mathrm{CD}}(\boldsymbol{\theta}, \bar{\boldsymbol{\theta}}) + \lambda \mathcal{L}_{\mathrm{GAN}}(\mathbf{f_\theta}, \mathcal{D}_\phi) \tag{7}$$

---

[1]A workaround exists when a pre-trained DM is not available, as investigated in recent works (Kong et al., 2024; Wang et al., 2024).

---

**Algorithm 1** The training pipeline for CM-GAN framework.

---

```
for iteration in range(number_of_iterations):
    σ = get_sigma(iteration)

    ## Train the discriminator
    for _ in range(k):
        real_img, t = sample_real_img(bs), sample_t(bs)
        noisy_img = diffusion_fwd_sampling(sample_real_img(bs), t)
        fake_img = f_θ(noisy_img, t)

        A = ψ_1(D_φ(gauss_blur(real_img,σ)), real_label)
        B = ψ_2(D_φ(gauss_blur(fake_img,σ)), fake_label)
        D_loss = -(A - B)
        compute_grad_and_update_D(D_loss, φ)

    ## Train the generator
    #  fake_img are fake images generated by f_θ to compute CM loss
    CM_loss, fake_img = get_CM_loss(real_img, sample_t(bs), f_θ)
    fake_img_blurred = gauss_blur(fake_img, blur_sigma)

    G_loss_disc = -ψ_2(D_φ(fake_img_blurred), real_label)
    G_loss = λ * G_loss_disc + CM_loss
    compute_grad_and_update_G(G_loss, θ)
```

---

where

$$\mathcal{L}_{\text{GAN}}(\mathbf{f}_{\boldsymbol{\theta}}, \mathcal{D}_{\boldsymbol{\phi}}) = \mathbb{E}[\psi_1(\mathcal{D}_{\boldsymbol{\phi}}(\mathbf{x}))] - \mathbb{E}\big[\psi_2\big(\mathcal{D}_{\boldsymbol{\phi}}(\mathbf{f}_{\boldsymbol{\theta}}(\mathbf{x}_{t_n}, t_n))\big)\big] \tag{8}$$

and $\mathbf{x} \sim p_{\text{data}}(\mathbf{x})$, $n \sim \mathcal{U}[\![1, N-1]\!]$, $\mathbf{x}_t \sim p_t(\mathbf{x}_t)$. Here, $\lambda$ is used to control the relative strength between the error correction signal from the discriminator $\mathcal{D}$ and the consistency constraints.

For values of $\lambda$ close to zero, the consistency loss becomes relatively stronger. This choice enhances training stability for the generator but limits its ability to refine outputs by incorporating error correction signals from the discriminator. Notably, when $\lambda = 0$, the setting simplifies to standard CM training. On the other hand, increasing $\lambda$ weakens the consistency constraints, giving the generator more flexibility to improve performance. However, this flexibility comes at the cost of reduced training stability and we could expect to observe similar syndromes occurring in the regular training of GAN. In Section 5, we demonstrate a sweet spot where an optimal balance between flexibility and stability is achieved, leading to improved performance of $\mathbf{f}_{\boldsymbol{\theta}}$.

At the early stages of training, the discriminator is often not sufficiently trained to provide meaningful feedback to the generator. To address this, it is beneficial to primarily guide the generator using consistency loss initially while gradually increasing the influence of the discriminator as its feedback becomes more reliable. One potential approach is to increase the weight parameter $\lambda$ from zero over time, but we found that continuously adjusting $\lambda$ can destabilize the training process. Instead, we apply a Gaussian blur with deviation $\sigma$ to all images used in GAN's training. The $\sigma$ is set to a large value initially and linearly decreases to zero as the training proceeds. This approach, inspired by Karras et al. (2021a) in their work on StyleGAN3-R, prevents the discriminator from focusing too early on high-frequency details, a strategy that also proves useful in our task. In addition, since the highly blurred real and fake images are largely not distinguishable, the discriminator will not provide effective signals to guide the generator, mimicking the effect of using a small $\lambda$ during the early training phase. Algorithm 1 summarizes the training pipeline of the proposed CM-GAN algorithm, while the sampling algorithm is identical to the one of the original CM model.

The proposed approach can also be viewed as a method for stabilizing GAN training, where the generator $\mathcal{G}$ adopts the same architecture as the CM's (Song et al., 2023). For a classical GAN's setting perspective,

the generator is defined as $\mathcal{G}(\boldsymbol{\epsilon}) = \mathbf{f}_\theta(\boldsymbol{\epsilon}, T)$ with $\boldsymbol{\epsilon} \sim \mathcal{N}(\mathbf{0}, T^2\mathbf{I})$, and is trained to fool the discriminator $\mathcal{D}_\phi$. In the CM-GAN setting, however, $\mathcal{G}$ is additionally trained to solve a sequence of auxiliary generation tasks. Conditioned on images corrupted by varying noise levels, the generator learns to produce realistic samples that can fool the discriminator. These auxiliary tasks can be interpreted as denoising tasks, where the main generation task corresponds to the limiting case where the input samples are too corrupted to retain any information about the original data. The generated samples are then regularized by consistency constraints in conjunction with a pre-trained PF-ODE model. Specifically, the pre-trained PF-ODE model serves as a prototype for $\mathcal{G}$, where the consistency constraint requires $\mathcal{G}(\boldsymbol{\epsilon})$ to be close to $\tilde{\mathbf{x}}_0(\boldsymbol{\epsilon})$—the starting point of the PF-ODE trajectory ending at $(\boldsymbol{\epsilon}, T)$. This prevents the generator from relying on a single plausible sample to fool the discriminator across all inputs $\boldsymbol{\epsilon}$. Instead, the generator is compelled to produce distinct and appropriate outputs for different $\boldsymbol{\epsilon}$, enhancing sample diversity and mitigating mode collapse. Moreover, the consistency constraints discourage the generator from blindly following the discriminator's error signal, reducing the likelihood of the generator being swayed by noisy or unstable feedback. This improves training stability and ensures that the generator produces more consistent and realistic samples.

**Applicability to the latent diffusion settings.** While we present our framework in pixel space, the same approach can be readily applied to latent space (Podell et al., 2023), where the adversarial objective can be implemented in either pixel space or latent space. Implementing the objective in pixel space allows practitioners to leverage existing (pretrained) discriminators and training techniques from numerous GAN-related works. However, this approach requires gradient backpropagation through the decoder, increasing training costs. In contrast, adversarial training in latent space eliminates this overhead, but it requires developing a custom discriminator for the latent code, potentially increasing implementation complexity. In our empirical study, we focus on the pixel-space setting, which provides sufficient evidence of the proposed method's effectiveness. We leave the extension to latent space for future work.

## 5 Empirical study

In this section, we empirically demonstrate the effectiveness of our CM-GAN framework and perform ablation studies to corroborate our previous discussions.

### 5.1 Experimental setups

**Datasets.** We conduct our empirical studies using two widely recognized benchmark datasets: IMAGENET 64×64 (Deng et al., 2009) and LSUN Bedroom 256×256 (Yu et al., 2015). IMAGENET 64×64 comprises over 14 million images across 1,000 object categories, while LSUN Bedroom contains three million high-resolution images showcasing diverse bedroom layouts and appearances.

**Experimental settings and implementations.** For the generator $\mathcal{G}$, we use the U-Net architecture from Song et al. (2023). To stabilize training, we apply an exponential moving average (EMA) to the generator's weights $\boldsymbol{\theta}$. The main model parameters $\boldsymbol{\theta}$ are optimized via stochastic gradient descent, while the EMA weights $\bar{\boldsymbol{\theta}}$ are updated as:

$$\bar{\boldsymbol{\theta}} \leftarrow \mu\,\bar{\boldsymbol{\theta}} + (1 - \mu)\,\boldsymbol{\theta} \quad \text{for} \ \ \mu \in [0, 1). \tag{9}$$

Following Song et al. (2023), we track EMA models with $\mu$ values of 0.999, 0.9999, and 0.999943219.

For adversarial training, we use the objective from Eq (7) and the discriminator from StyleGAN-XL (Sauer et al., 2022), applying the hinge loss (Lim & Ye, 2017) for $\mathcal{L}_{\text{GAN}}$. Specifically,

$$\psi_1\big(\mathcal{D}_\phi(\mathbf{y})\big) = -\max\big(0, 1 - \mathcal{D}_\phi(\mathbf{y})\big),$$
$$\psi_2\big(\mathcal{D}_\phi(\mathbf{y})\big) = \max\big(0, 1 + \mathcal{D}_\phi(\mathbf{y})\big).$$

Unless otherwise noted, we set $\lambda$ in Eq (7) to $10^{-5}$ (see Section 5.2 for details on its selection). We also replace the quadratic loss in $\mathcal{L}_{\text{CD}}$ with LPIPS (Zhang et al., 2018), as it significantly improves performance in classical CM training (Song et al., 2023). We train the generator following Algorithm 1 with $k = 1$.

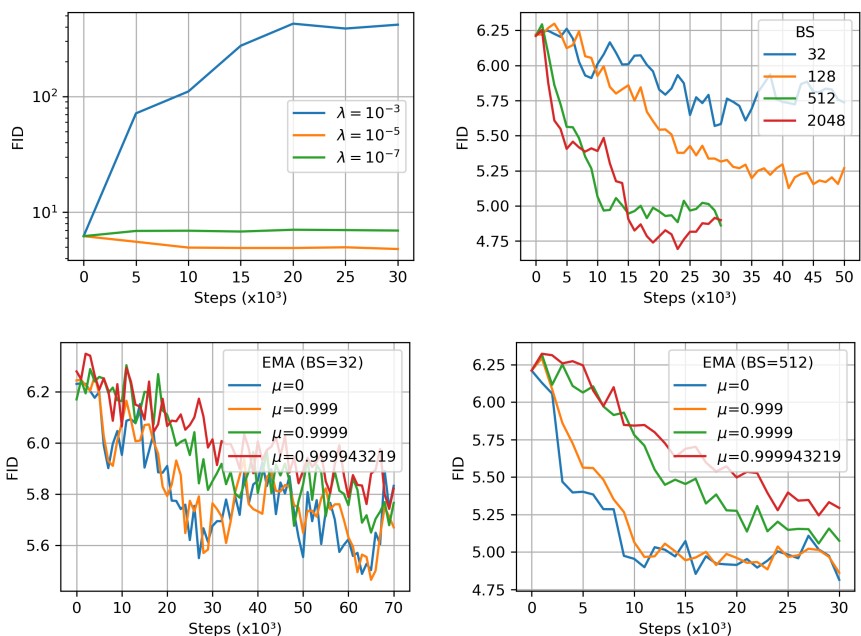

Figure 2: Various factors that affect the generated image quality in FID on ImageNet 64 x 64.

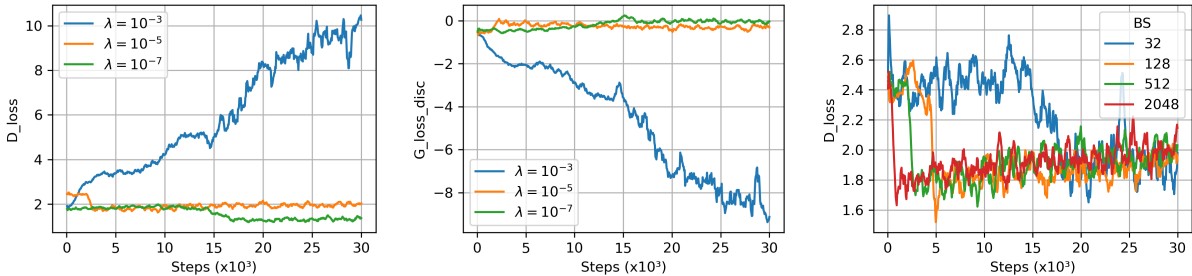

Figure 3: The training loss trajectories for different training settings on ImageNet 64 x 64. D_loss and G_loss_disc, defined in Algorithm 1, are empirical estimators of $-\mathcal{L}_{\mathrm{GAN}}(\mathbf{f}_{\boldsymbol{\theta}}, \mathcal{D}_{\boldsymbol{\phi}})$ and $-\mathbb{E}\left[\psi_2\left(\mathcal{D}_{\boldsymbol{\phi}}(\mathbf{f}_{\boldsymbol{\theta}}(\mathbf{x}_{t_n}, t_n)))\right)\right]$ in Eq (8).

**Other settings** For the Gaussian blurring schedule in GAN training, we set the initial blur parameter of $\sigma = 10$ and linearly reduce it to zero over the first 300K images. Unless otherwise noted, all models are optimized using the Rectified Adam optimizer (Liu et al., 2020), with a learning rate of $10^{-7}$ and a batch size of 512 for ImageNet 64×64, and a learning rate of $10^{-6}$ with a batch size of 48 for the Bedroom 256×256. The generator's weights, along with their EMA counterparts, are initialized from the checkpoints provided by Song et al. (2023), while the discriminator's weights are initialized from the checkpoints used in StyleGAN-XL (Sauer et al., 2022). Unless specified otherwise, we report results based on the EMA model with a decay factor of $\mu = 0.999$. All experiments were conducted on 4 Nvidia L40S GPUs.

## 5.2 CM-GAN stabilizes the training of GAN

In Section 4, we noted that CM-GAN is expected to stabilize the training process, with an optimal value for $\lambda$ (as defined in Eq (7)). The first plot in Fig 2 shows the Fréchet Inception Distance (FID, Heusel et al. 2017) for various $\lambda$ values. FID scores, calculated from 50K generated images, indicate higher image fidelity with lower values.

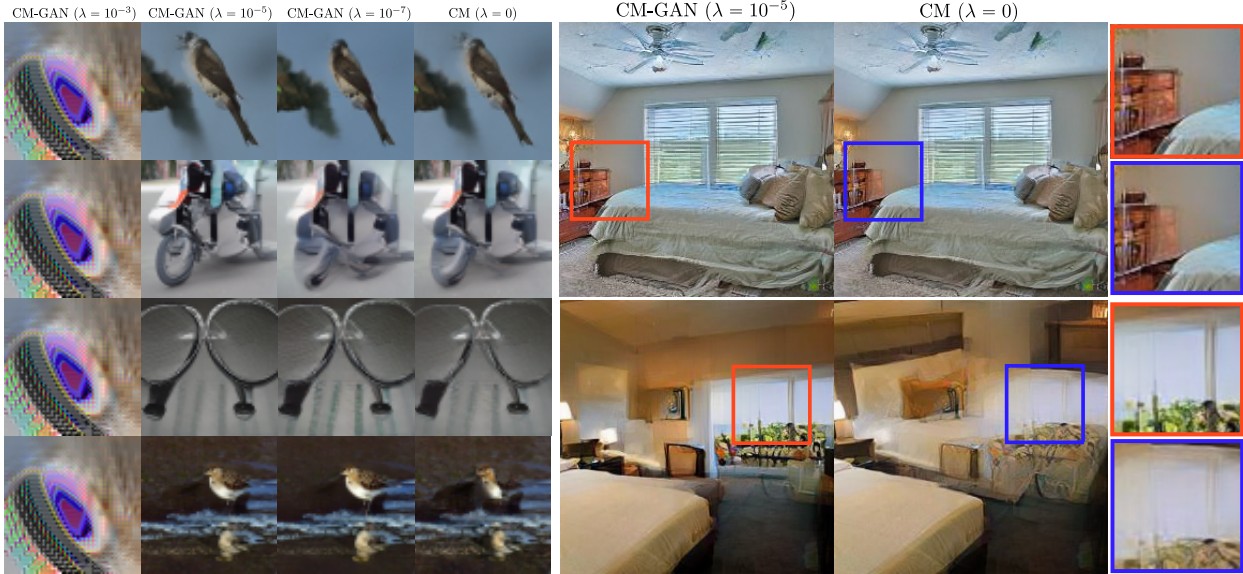

Figure 4: Images sampled by models trained on ImageNet 64×64 (left) and Bedroom 256×256 (right). (NFE=1)

Table 1: Performance comparison (mean ± std) on ImageNet 64 × 64 and LSUN Bedroom 256 × 256. NFE refers to the number of function evaluations. The standard deviations (std) are computed from five independent sampling runs with different random seeds. Lower FID indicates better sample quality, while higher Precision and Recall reflect better sample fidelity and diversity, respectively.

| METHOD | NFE (↓) | FID (↓) | Prec. (↑) | Rec. (↑) |
|---|---|---|---|---|
| **ImageNet 64 × 64** | | | | |
| PD (Salimans & Ho, 2022) | 1 | 15.39 | 0.59 | 0.62 |
| DFNO (Zheng et al., 2023) | 1 | 8.35 | - | - |
| CM (Song et al., 2023) | 1 | 6.20 | **0.68** | **0.63** |
| CMGAN | 1 | $4.69 \pm 0.035$ | $\textbf{0.68} \pm 0.015$ | $\textbf{0.63} \pm 0.017$ |
| PD (Salimans & Ho, 2022) | 2 | 8.95 | 0.63 | **0.65** |
| CM (Song et al., 2023) | 2 | 4.70 | 0.69 | 0.64 |
| CMGAN | 2 | $\textbf{3.68} \pm 0.038$ | $\textbf{0.71} \pm 0.001$ | $\textbf{0.65} \pm 0.002$ |
| ADM (Dhariwal & Nichol, 2021) | 250 | **2.07** | 0.74 | 0.63 |
| EDM (Karras et al., 2022) | 79 | 2.44 | 0.71 | **0.67** |
| BigGAN-deep (Brock et al., 2019) | 1 | 4.06 | **0.79** | 0.48 |

| METHOD | NFE (↓) | FID (↓) | Prec. (↑) | Rec. (↑) |
|---|---|---|---|---|
| **LSUN Bedroom 256 × 256** | | | | |
| PD (Salimans & Ho, 2022) | 1 | 16.92 | 0.47 | 0.27 |
| CM (Song et al., 2023) | 1 | 7.80 | **0.66** | 0.34 |
| CMGAN | 1 | $\textbf{6.23} \pm 0.010$ | $0.65 \pm 0.001$ | $\textbf{0.39} \pm 0.002$ |
| PD (Salimans & Ho, 2022) | 2 | 8.47 | 0.56 | 0.39 |
| CM (Song et al., 2023) | 2 | 5.22 | **0.68** | 0.39 |
| CMGAN | 2 | $\textbf{4.79} \pm 0.011$ | $\textbf{0.68} \pm 0.002$ | $\textbf{0.41} \pm 0.001$ |
| DDPM (Ho et al., 2020) | 1000 | 4.89 | 0.60 | 0.45 |
| ADM (Dhariwal & Nichol, 2021) | 1000 | **1.90** | **0.66** | **0.51** |
| EDM (Karras et al., 2022) | 79 | 3.57 | **0.66** | 0.45 |
| PGGAN (Karras et al., 2018) | 1 | 8.34 | - | - |
| StyleGan2 (Karras et al., 2020) | 1 | 2.35 | 0.59 | 0.48 |

Starting with a well-trained CM model, we observe that CM-GAN improves sample quality when $\lambda = 10^{-5}$. However, at $\lambda = 10^{-3}$, the influence of CM constraints weakens, making the training resemble classical GAN

behaviour and becoming less stable. This instability is reflected in a sharp rise in FID, indicating that the GAN component disrupts the generator's learned features. To understand this, the first two plots in Fig 3 show the trajectories of D_loss and G_loss_disc (defined in Algorithm 1). A rapid increase in D_loss and a decrease in G_loss_disc signal mode collapse, where the discriminator is stuck in a local minimum, and the generator repeatedly produces the same image to fool the discriminator. This collapse is further illustrated in the first column of Fig 4, where the nearly identical outputs for different input $\mathbf{x}_T \sim \mathcal{N}(\mathbf{0}, T^2\mathbf{I})$ confirm our observations. On the other hand, when $\lambda$ is reduced to $10^{-7}$, the discriminator's signal becomes too weak to guide the generator effectively. While the loss trajectories in Fig 3 indicate stable training, the FID increases slightly, as shown in Fig 2. This FID increase could be due to the smaller batch size in our experiments, which results in less stable training gradients than the original CM training (we used a batch size of 512, while the original CM training used 2048).

## 5.3 Batch size selections and EMA

While the second plot in Fig 2 shows that a larger batch size accelerates the FID drop and improves image quality, the FID also decreases steadily even with a batch size as small as 32. This indicates that CM-GAN can still enhance generator performance with smaller batch sizes and a few training iterations, even when GPU memory is limited. However, when memory is sufficient, opting for a larger batch size is recommended for optimal results. Importantly, as shown in the last plot of Fig 3, training stability is not significantly impacted by the batch size, provided that $\lambda$ is appropriately selected.

In the last two plots of Fig 2, we visualize the FID trajectory for different weight decay values $\mu$ used in EMA with batch sizes of 32 and 512. ($\mu = 0$ corresponds to the main model without applying EMA.) For small batch sizes, a $\mu$ value close to one effectively stabilizes the FID trajectory, though at the cost of slower convergence. In contrast, with larger batch sizes, the stability gain from increasing $\mu$ is marginal, making the cost of slower convergence less worthwhile.

## 5.4 CM-GAN enhances CM's performance

In Section 4, we discussed how CM-GAN can be viewed as a fine-tuning method to enhance CM models by guiding them toward the true data distribution through the discriminator. Fig 4 presents the one-step outputs of CM-GAN generators for different values of $\lambda$. (Additional samples can be found in the supplementary materials.) Images in the same row are generated using the same input $\mathbf{x}_T \sim \mathcal{N}(\mathbf{0}, T^2\mathbf{I})$. When $\lambda = 0$, the framework reduces to the original CM, and we use checkpoints from Song et al. (2023). As shown, with a proper $\lambda = 10^{-5}$, CM-GAN effectively corrects abnormalities in CM-generated images while preserving global features. This preservation is enforced by the consistency constraints, which also stabilize the adversarial training process.

Table 1 presents the FID scores of CM-GAN and other generative models on ImageNet 64×64 and Bedroom 256×256, with a training batch size of 2048 for the ImageNet and 48 for the Bedroom. The results are divided into three categories: distillation models with one and two function evaluations (NFE), and generative models trained directly from the dataset. Compared to standard CMs, the significantly lower FID scores achieved by CM-GAN highlight its effectiveness in improving the performance of CM models across both datasets.

## 5.5 Training from scratch or finetuning

While CM-GAN shows its effectiveness in improving trained CM models, Fig 5 shows that it does not accelerate the FID drop during the early stages of training. Notably, adversarial training typically requires a lower learning rate for additional stability. For instance, in our experiments on ImageNet, we set the learning rate to $10^{-7}$, whereas the original CM uses a learning rate of $8 \times 10^{-6}$, leading to an even faster FID drop than shown in Fig 5. Therefore, we recommend using regular CM training in the initial stages and applying CM-GAN in the final phase to maximize model performance.

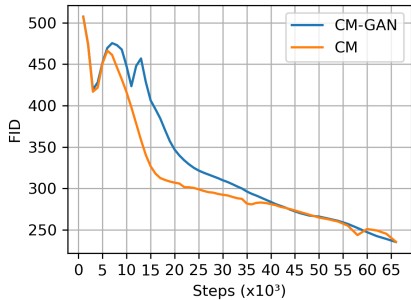

Figure 5: FID trajectories of CM and CM-GAN when training from scratch on ImageNet 64×64.

## 5.6 Adversarial objective corrects diffusion model training flaws

To provide more convincing evidence that the adversarial objective can effectively correct estimation errors in diffusion models (DM) caused by imperfect training, we applied CM-GAN training using an artificially corrupted diffusion model for distillation.

To create the corrupted diffusion model, we started with the diffusion model pretrained on ImageNet $64 \times 64$ from the official CM repository (Song et al., 2023) and corrupted the weights of the last convolutional layer by adding Gaussian noise with a standard deviation of 0.01. After corruption, the diffusion model achieved an FID of 20.77, which serves as the baseline.

Next, we distilled a CM model from the corrupted DM, which achieved an FID of 23.41 using one-step sampling. To improve its performance, we applied CM-GAN training by introducing an adversarial objective starting from the trained CM model. This reduced

Table 2: FID (mean ± std) evolution when distilling from a corrupted DM.

| Model | FID |
|---|---|
| Corrupted DM | $20.77 \pm 0.042$ |
| CM | $23.41 \pm 0.031$ |
| CM-GAN | $\mathbf{10.38} \pm 0.017$ |

the FID to 10.38. All models were trained with a batch size of 256 and a learning rate of $8 \times 10^{-6}$. For CM-GAN, we set $\lambda = 10^{-5}$. The FID results are summarized in Table 2.

As shown in Table 2, adding the adversarial training objective enables the CM model to surpass the baseline DM's performance. This suggests that the GAN structure can effectively correct imperfections in the original DM during the distillation process.

## 6 Discussion

In this paper, we introduced CM-GAN, a technique that enhances GAN training stability while also serving as an effective finetuning method for CMs. Our empirical study on standard benchmark datasets demonstrates that the consistency constraint acts as a powerful regularizer, stabilizing GAN training and helping to prevent mode collapse. Furthermore, when used as a finetuning method, CM-GAN significantly improves the sample quality of CM models, even under GPU resource limitations that necessitate small batch sizes and fewer training iterations.

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

## A Appendix

In this section, we present additional samples generated by models trained on the ImageNet 64×64 and Bedroom 256×256 datasets using the CM-GAN framework, with varying values of $\lambda$. As in the main text, all images are produced using one-step sampling (NFE = 1). Each row contains images generated from the same initial condition, $\mathbf{x}_T \sim \mathcal{N}(\mathbf{0}, T^2\mathbf{I})$. When $\lambda = 0$, the framework becomes the original CM, and we use the pretrained checkpoints from Song et al. (2023).

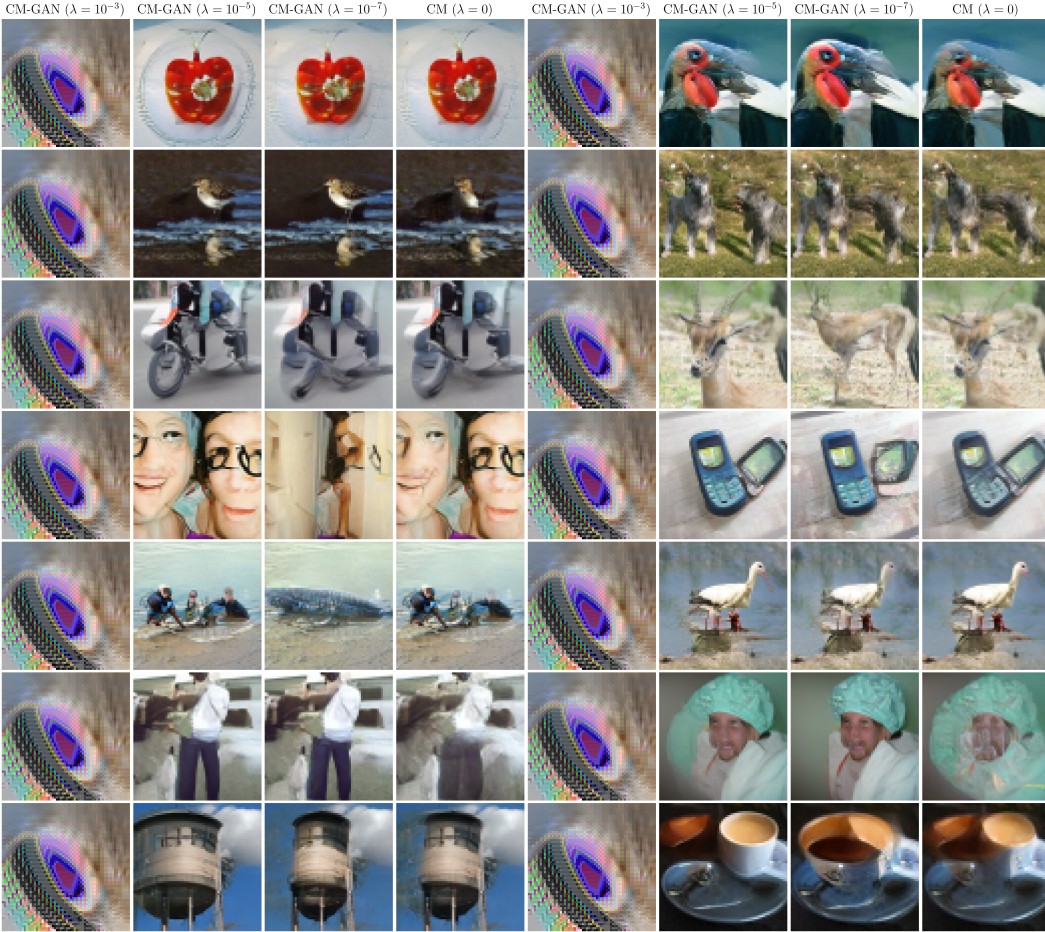

Figure 6: ImageNet $64 \times 64$

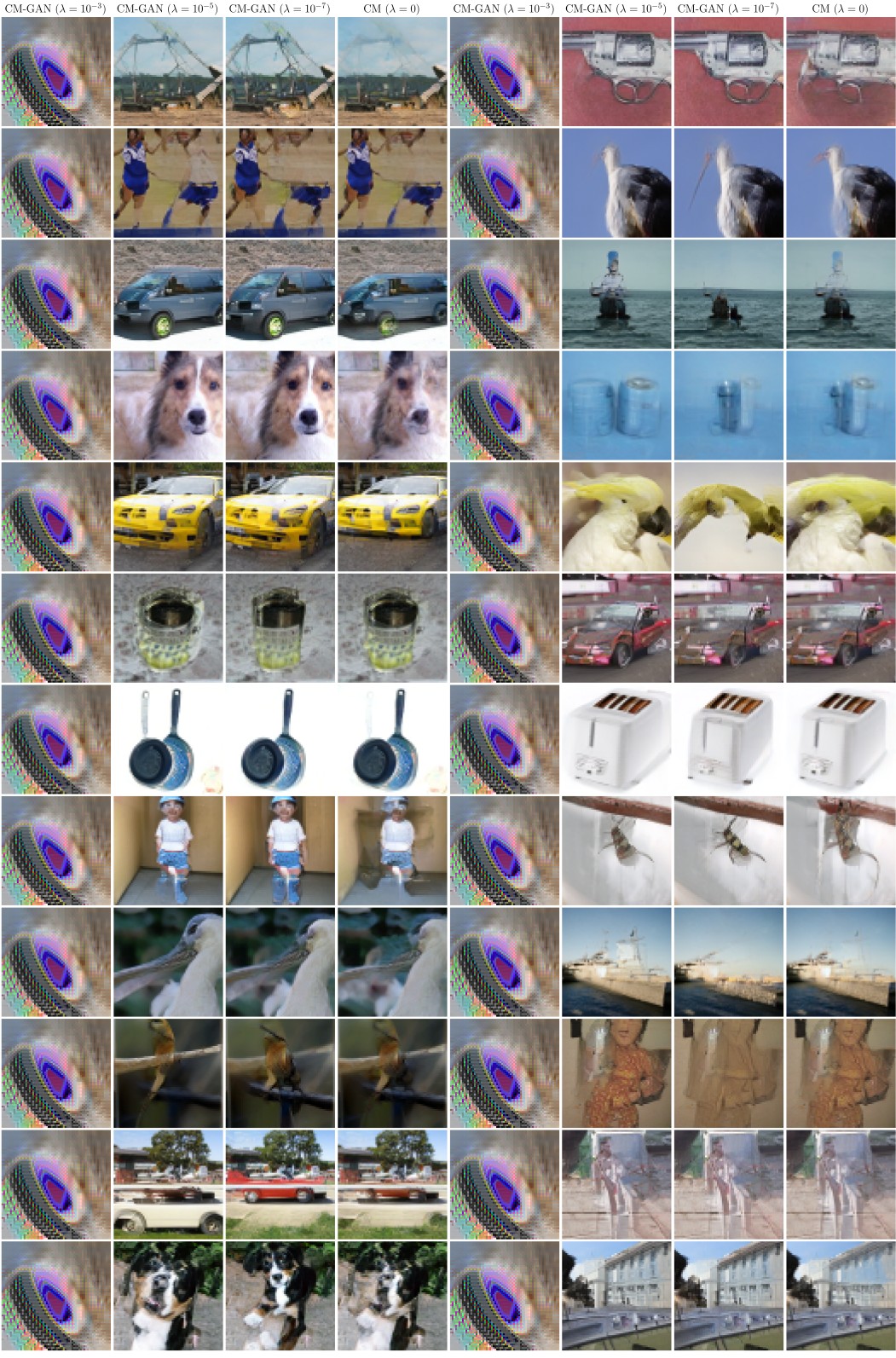

Figure 7: ImageNet $64 \times 64$ (continued)

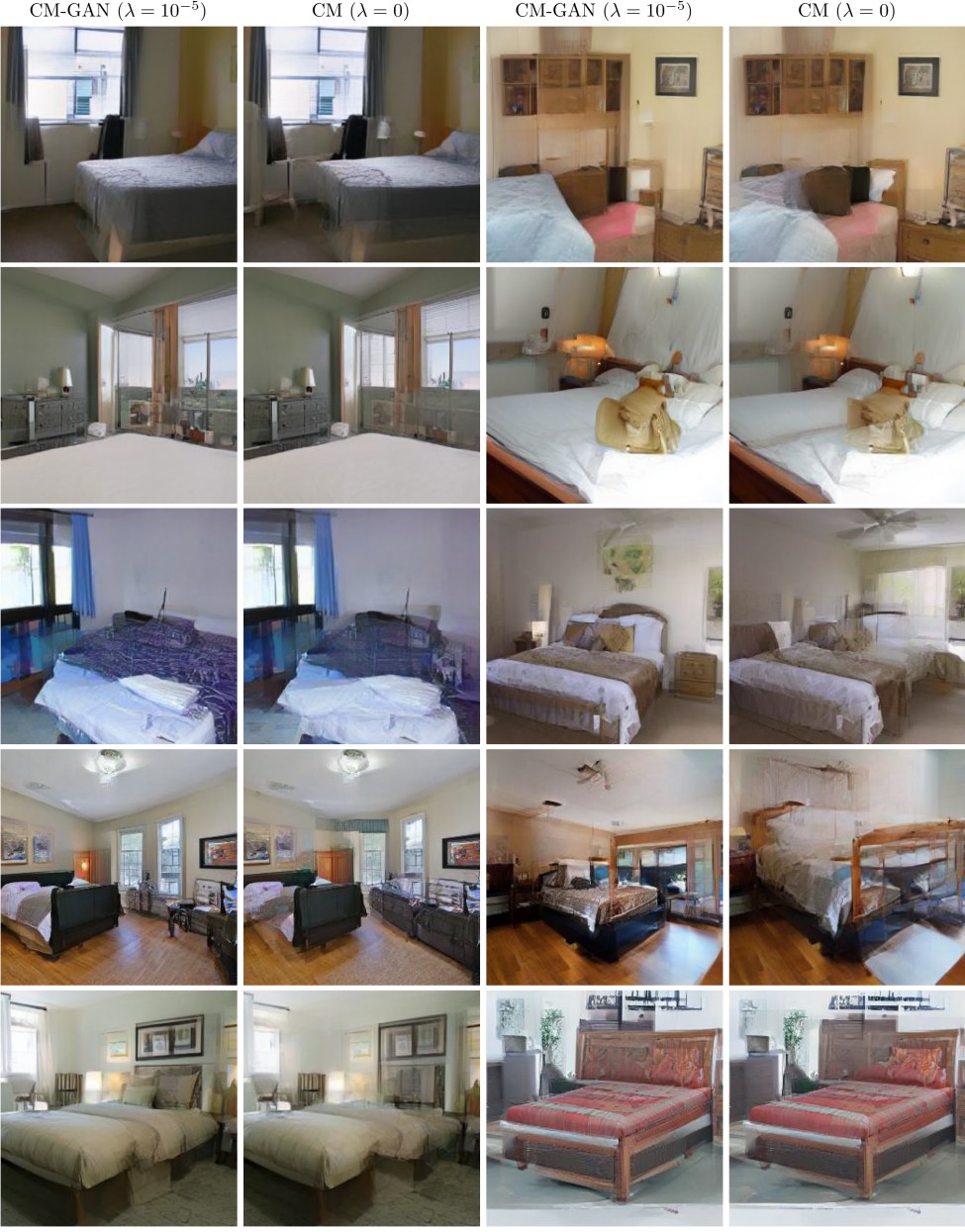

Figure 8: Bedroom 256×256

CM-GAN ($\lambda = 10^{-5}$)    CM ($\lambda = 0$)    CM-GAN ($\lambda = 10^{-5}$)    CM ($\lambda = 0$)

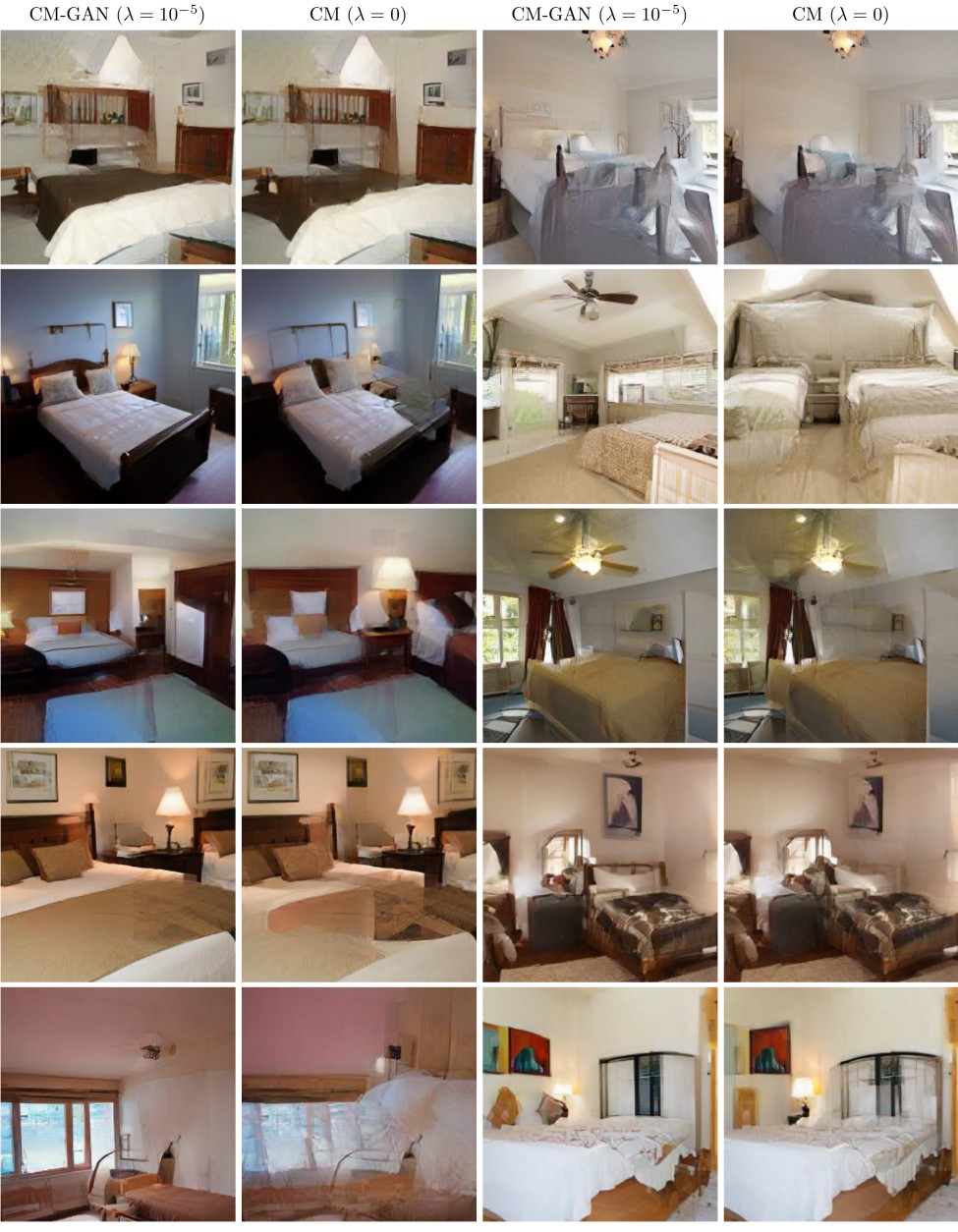

Figure 9: Bedroom 256×256 (continued)

