# OpenReview forum: "CM-GAN: Enhancing Consistency Model Image Quality and Stabilizing GAN Training"
_TMLR — Rejected by TMLR_

### Review · Reviewer_THib · 2025-01-08

**Summary Of Contributions:**

This paper studies the image generation problem by combining methodologies from diffusion model-based approaches and generative adversarial networks (GAN). The key idea in this paper is to replace the L2 loss used to train the consistency model (CM) in diffusion-based generation with a discriminator-based adversarial loss and LPIPS loss. The authors show that for the purpose of training a better GAN model, the proposed method can leverage a diffusion model as the prior to provide better stability,  compared to using pure adversarial loss. The method also potentially leads to a better finetuning method for CMs, to facilitate diffusion model distillation and acceleration.

**Audience:**

Yes

**Broader Impact Concerns:**

There is no ethical concerns for this work.

**Claims And Evidence:**

No

**Requested Changes:**

1. The authors stated that "This performance gap (between GANs and diffusion models) can be attributed to the intricate GAN-specific architectures necessary for maintaining stability in adversarial training." without giving a reference. Since this might not be widely accepted, the authors should either give more evidence or provide a reference for it.
2. For the statement "we do not have a similar method to classifier-free guidance", the authors should explain why the classifier-free guidance is important for GAN. Do you mean the discriminator in GAN make it not classifier-free?
3. Introduction to SDS in the following paragraph could be omitted.
4. The authors might want to give a more intuitive example for $\psi_1$ and  $\psi_2$ in the sentence following Eq. (3).
5. Fig. 1 is hard to understand. Maybe adding a title for the vertical axis can help.

**Strengths And Weaknesses:**

# Strength

1. The aim of this study is to accelerate the diffusion-based image generative model by avoiding the iterative denoising. This is indeed an important topic and what is discussed in the paper might provide some insights.

2. The idea, i.e. training CM with an adversarial loss, is somehow interesting and intuitive. Experimental results clearly demonstrate that the proposed method improves training stability.

# Weakness

1. [Clarity of Contribution] The introduction suggests that the proposed method can benefit diffusion models with the following sentence "Additionally, it acts as a fine-tuning mechanism for CMs by integrating a discriminator, potentially exceeding the performance of standard CMs." However, the authors didn't provide any evidence, even preliminary, to support this claim. There could be potentially an over-claiming issue.

2. [Visual Quality]. As a paper studying image generation, the qualitative results shown in the paper is far off in quality compared to existing works, with either GAN models or diffusion models. Yet there is no evidence in this paper that the proposed method can potentially facilitate existing image generation approaches.

3. The proposed method still relies on pretrained CMs. It somehow restricts the applicability.

4. Although the experimental results mostly support the main claim, the authors did not compared the proposed method to any other existing ones, making the claim less strong.

---

> ### Author Response · Authors · 2025-03-24
> **Thank you for your comments!**
>
> Thank you for taking the time to review our paper and provide valuable feedback. Please find our responses to your comments below:
>
>
> **Q1.** Regarding the performance gap between GANs and diffusion models
>
> **A1.** We appreciate the reviewer’s observation and acknowledge that this statement may require additional empirical support. Our intent was to convey that the inherent instability of adversarial training often necessitates more specialized network architecture designs to improve training stability. However, these architectural constraints can limit the model’s expressiveness, which may, in turn, negatively affect performance. In the revised version, we have softened the wording and added more references to better support and clarify this statement.
>
> **Q2.** For the statement "we do not have a similar method to classifier-free guidance", the authors should explain why the classifier-free guidance is important for GAN. Do you mean the discriminator in GAN make it not classifier-free?
>
> **A2.** Classifier-free guidance is a sampling technique used in diffusion models that extrapolates the conditioned and unconditioned score functions learned during training. Diffusion models often outperform typical GAN-based implementations even without classifier-free guidance. However, this post-training method can further enhance the already strong performance of diffusion models. Since it relies on score-based continuous sampling—a method exclusive to diffusion models—it is not directly applicable to GANs due to their fundamentally different sampling methods.
>
> We have rephrased the relevant sentences in the revised version to clarify our message.
>
> **Q3.**  Introduction to SDS in the following paragraph could be omitted.
>
> **A3.** We have shortened the related discussion but chose not to remove it entirely, as the technique is adopted in the works related to our submission, which we discuss in the subsequent paragraphs.
>
> **Q4.**  The authors might want to give a more intuitive example for $\psi_1$  and  $\psi_2$ in the sentence following Eq. (3).
>
> **A4.** In the revised version, we have clarified the meaning of $\psi_1$ and $\psi_2$ in the context of the classical GAN formulation.
>
> **Q5.** Fig. 1 is hard to understand. Maybe adding a title for the vertical axis can help.
>
> **A5.** We have replotted the figure and added labels to both the x and y axes as suggested.
>
> **Q6.** Regarding Clarity of Contribution and specifically the sentence "*Additionally, it acts as a fine-tuning mechanism for CMs by integrating a discriminator, potentially exceeding the performance of standard CMs.*"
>
> **A6.** To support the claim that the proposed method can be considered a fine-tuning mechanism, we have included performance results with and without adversarial objectives in Table 1. (To prevent potential confusion, we have renamed the model from CD to CM in the revised version; CD was used in the original Consistency Models paper to refer to consistency models distilled from a pretrained diffusion model.)

---

### Review · Reviewer_aMSg · 2025-02-06

**Summary Of Contributions:**

This paper proposes CM-GAN, a method that leverages adversarial training to enhance the sample quality of consistency models (CMs).  The key idea is to combine a discriminator from GANs with a consistency constraint from CMs, thereby leveraging the strengths of both models while mitigating their respective limitations.  The proposed approach leads to improved image quality and training stability, as demonstrated through empirical results on benchmarks (ImageNet and Bedroom datasets).

**Audience:**

Yes

**Broader Impact Concerns:**

No broader impact concerns.

**Claims And Evidence:**

No

**Requested Changes:**

The main issue with the paper is that it's contribution is not clear. For example, Consistency Trajectory Models propose also an adversarial distillation training scheme. It is not clear how the proposed solution refers with respect to this one? The authors list and briefly discuss CTM in a paragraph "Concurrently, ..." but this method seems to have been published more than an year before the submission of this paper.  I would recommend the authors to do a more exhaustive comparison about the similarities and differences with respect to this an possible other work doing Adversarial consistency distillation.

Additionally, I recommend the authors to incorporate a discussion about the applicability of this method to Latent Diffusion Models, which are currently the state-of-the-art  text-to-image generation frameworks.

Incorporate a better discussion about the claim that the diffusion model doesn't need to be fully-trained (could be imperfect or undertrained as stated by the authors), and the adversarial loss will help to improve the quality. If this is the case we should be able to see results where the distilled model outperforms the baseline. Consider adding experiments on this regard.

There's another claim that sort of contradicts this, which is the sentence that says that "fast sampling of CMs comes with a trade-off in output quality since the pretrained PF-ODE model cannot be perfectly distilled in general". This is in some sort of contradiction to the claim that with the adversarial training you can improve the quality of the imperfect baseline.

After Eq(6). Please give a mathematical definition of $\hat{\textbf{x}}_{t_n}$ (maybe as an example use Euler's update).

Unclear sentence (page 6). "Apart from the main generation task, $\mathcal{G}$ is also trained to compalte a sequence of auxiliary denoising tasks...". I couldn't understand what this sentence (and the paragraph) is trying to explain. Please reformulate.

**Strengths And Weaknesses:**

Strengths:
  - The paper is sound, the adopted solution is reasonable and shows an improvement on the generation quality
  - The paper is overall well-written with enough detail to reproduce it.

Weaknesses:
 - The paper doesn't introduce novel ideas (or at least the novelty with respect to previous work is not clear, e.g., CTM by Kim et al. [2024], appear on arxiv on Oct 2023, published at ICLR 2024).
 - The evaluation is limited. For example, there's no discussion and experimentation on Latent Diffusion Models.
 - Some secondary claims are not really validated. For example,  if the starting model is undertrained, adversarial training will help to improve it (see more info below in Requested Changes).
 - (Minor) A few points that require clarifications that I list Request Changes.

---

> ### Author Response · Authors · 2025-03-24
> **Thank you for your comments!**
>
> Thank you for reviewing our submission and valuable comments. Here are our responses:
>
> **Q1.** Regarding the novelty and contribution.
>
> **A1.** We are confident that our work makes novel contributions. However, providing sufficient details in the original submission and responses could risk violating the double-blind policy. We are consulting the AE for guidance and appreciate your understanding.
>
> **Q2.** I recommend the authors to incorporate a discussion about the applicability of this method to Latent Diffusion Models, which are currently the state-of-the-art text-to-image generation frameworks.
>
> **A2.** In the revised version, we have added a paragraph at the end of Section 4 discussing the applicability of the proposed method to latent diffusion models. Since the empirical study in pixel space has already provided strong evidence of the framework’s effectiveness and we have limited computational resources to train or fine-tune latent diffusion models, we plan to defer the related empirical study to future work.
>
> **Q3.** Incorporate a better discussion about the claim that the diffusion model doesn't need to be fully-trained (could be imperfect or undertrained as stated by the authors), and the adversarial loss will help to improve the quality ...
>
> **A3.**  We have added Section 5.6 in the revised version to provide more convincing empirical results and additional discussion to support this claim.
>
>
> **Q4.** There's another claim that sort of contradicts this, which is the sentence that says that "fast sampling of CMs comes with a trade-off in output quality since the pretrained PF-ODE model cannot be perfectly distilled in general". This is in some sort of contradiction to the claim that with the adversarial training you can improve the quality of the imperfect baseline.
>
> **A5.** We do not see an obvious inconsistency here. In general, the imperfect training of the distilled CM model can be attributed to three main factors: (1) the diffusion model, which serves as the teacher, may not be perfectly trained; (2) the distillation process involves evaluating the PF-ODE, which introduces discretization errors; and (3) the CM neural network may not be perfectly trained through gradient descent. The last two factors explain why the pretrained PF-ODE model cannot be perfectly distilled. However, adversarial training directly guides the CM to produce more realistic samples, which can help mitigate the training imperfections caused by these factors. We are happy to provide further clarification if needed.
>
>
> **Q5.** After Eq(6). Please give a mathematical definition of $\hat{\mathbf{x}}_{t_n}$ (maybe as an example use Euler's update).
>
> **A5.** We have incorporated the definition as advised.
>
> **Q6.** Unclear sentence (page 6). "Apart from the main generation task, $\mathcal{G}$ is also trained to complete a sequence of auxiliary denoising tasks...". I couldn't understand what this sentence (and the paragraph) is trying to explain. Please reformulate.
>
> **A6.** We have rewritten the paragraph to more clearly explain why CM-GAN can be viewed as a method for stabilizing GAN training.

---

> > ### Comment · Reviewer_aMSg · 2025-04-16
> > **Novelty.**
> >
> > *Q1. Regarding the novelty and contribution.*
> >
> > *A1. We are confident that our work makes novel contributions. However, providing sufficient details in the original submission and responses could risk violating the double-blind policy. We are consulting the AE for guidance and appreciate your understanding.*
> >
> > Do the authors have an update on this regard?

---

> > > ### Author Response · Authors · 2025-04-16
> > >
> > > Dear Reviewer aMSg,
> > >
> > > While we have communicated with the AE to clarify in detail how our work relates to the existing literature, particularly CTM, the AE has asked us to adhere strictly to the double-blind policy during the rebuttal process. As a result, we are unable to provide further information on this matter at this time. Given that the AE is likely aware of the relevant context and will be making the final decision, we would greatly appreciate it if an additional comment could be provided indicating what the recommendation would be in the absence of originality concerns when making the final decision.
> > >
> > > Sincerely,
> > >
> > > Authors.

---

### Review · Reviewer_bGvt · 2025-03-14

**Summary Of Contributions:**

The paper presents a study of CM-GAN, a framework combining consistency models and GAN training. The paper can be succintly described as a variation of ADD https://arxiv.org/pdf/2311.17042 replacing the Diffusion model with a consistency model and simplifying by removing prompting with natural text. Evaluations are done on imagenet 64 and lsun 256 and present the result model favourably against similar methods.

**Audience:**

Yes

**Broader Impact Concerns:**

not required

**Claims And Evidence:**

Yes

**Requested Changes:**

1. To compare the models, I would request the authors follow https://arxiv.org/abs/1811.12808 to get statisttically meaningful comparisons => critical
2. consider discussing the following references (some might be in the references already if I misssed them, but a quick read noted them as absent and some might be relevant => not critical, feel free to discard irrelevant ones

https://arxiv.org/abs/2405.05967
https://ojs.aaai.org/index.php/AAAI/article/view/29503
https://arxiv.org/abs/2311.14097v3

**Strengths And Weaknesses:**

- strengths: I think the paper is well written in that it presents the core background efficiently and then clearly explains the main idea to the reader, the authors give suitable context (some nice-to have citations I did not pick up on are in the requested changes, but nothing critically is missing) and the method appears to give an improvement based on the benchmarks presented. I appreciate the ablations performed as well

- weaknesses:

1. evaluation only uses two datasets, and small ones (but acceptable I think, for computational reasons)
2. while the 50k images should reduce variance by quite a lot, error bars and even better, statistical significance (maybe with bootstrap,s maybe with more samples) should be estimated => requested change
3. compared to e.g. ADD less featureful

---

> ### Author Response · Authors · 2025-03-24
> **Thank you for your comments!**
>
> Thank you very much for reviewing our paper and for the constructive feedback. Below are our responses to your comments:
>
> **Q1**. Regarding the statistically meaningful comparisons of model performance
>
> **A1**. We acknowledge that the results could be more convincing with statistical significance tests. However, we would like to clarify that such methods are not commonly used in the evaluation of diffusion models. In this paper, we followed the standard evaluation protocols used in existing generative model research, such as EDM [1] and Consistency Models [2].
>
> To address this concern, we have updated Table 1 to include the standard deviations, calculated from five independently sampled sets of 50K images generated with different random seeds.
>
> [1] Tero Karras, Miika Aittala, Timo Aila, Samuli Laine. Elucidating the Design Space of Diffusion-Based Generative Models. NeurIPS 2022.
>
> [2] Yang Song, Liyue Shen, Lei Xing, Stefano Ermon. Consistency Models. ICML 2023.
>
> **Q2**. Regarding the suggested references
>
> **A2**. We agree that all three suggested references are relevant and have incorporated them into the revised version of the paper.

---

### Decision · Action_Editor_CN8U · 2025-04-19

**Recommendation:** Reject

**Comment:**

Although the author provided revision and rebuttal to the reviewers, two of the reviewers (THib and aMSg) gave leaning reject and reject scores in the final stage. The reviewer bGvt stated that it would be better to show a statistical significance test using public checkpoints, but bGvt noted that the paper has reasonable analysis. The main issue raised by THib is the image quality, and aMSg pointed out the missing analysis and comparison with similar approaches.

The authors privately contacted AE about the comparison with a similar approach, since they worried it may violate the double-blind policy. AE replied to the authors, stating that the authors should keep the double-blind policy and respond to the reviewer from a third-person point of view. However, the authors replied to the reviewer as follows:

> While we have communicated with the AE to clarify in detail how our work relates to the existing literature, particularly CTM, the AE has asked us to adhere strictly to the double-blind policy during the rebuttal process. As a result, we are unable to provide further information on this matter at this time.

This is misleading since AE did not restrict authors from clarifying the difference between this work and their prior work. As stated by aMSg, comparison with CTM is crucial since it also employs an adversarial distillation training scheme.

As a final decision, accepting the paper in its current form cannot resolve the concerns raised by reviewers. AE cannot recommend the acceptance of this paper. The authors are advised to submit the paper after resolving the issues.

**Audience:**

Effective dataset generation techniques receive attention from the field since they can effectively address the lack of original datasets. In addition, stable training of the generative models is essential for studying.

**Claims And Evidence:**

The paper proposes CM-GAN, a framework that enhances image generation by integrating consistency models (CMs) with adversarial training from GANs. It replaces the traditional L2 loss in consistency models with a discriminator-based adversarial loss and LPIPS loss, aiming to combine the strengths of GANs and CMs for improved image quality and training stability. Compared to related work like ADD, CM-GAN simplifies the architecture by removing text prompts and diffusion models. Experiments on ImageNet 64 and LSUN 256 (Bedroom) suggest competitive performance.

However, the paper faces concerns regarding a lack of clarity on its novelty. Notably, it does not sufficiently differentiate itself from Consistency Trajectory Models (CTM)—a prior method that also employs adversarial distillation for consistency training. The discussion of CTM is brief and lacks a substantive comparison, leaving the paper's unique contributions ambiguous. AE agrees on that A more comprehensive analysis of related work and a clearer articulation of CM-GAN’s distinct value are necessary for the work to be considered for acceptance.

**Resubmission Of Major Revision:**

The authors may consider submitting a major revision at a later time.